# Changes in Pronoun Use a Decade before Clinical Diagnosis of Alzheimer’s Dementia—Linguistic Contexts Suggest Problems in Perspective-Taking

**DOI:** 10.3390/brainsci12010121

**Published:** 2022-01-17

**Authors:** Dagmar Bittner, Claudia Frankenberg, Johannes Schröder

**Affiliations:** 1Leibniz-Centre General Linguistics, 10117 Berlin, Germany; 2Section of Geriatric Psychiatry, Department of Psychiatry, University of Heidelberg, 69115 Heidelberg, Germany; Claudia.Frankenberg@med.uni-heidelberg.de

**Keywords:** prodromal Alzheimer’s Dementia, pronoun use, evaluative and elaborative information, German

## Abstract

The use of pronouns has been shown to change pathologically in the early phases of Alzheimer’s Dementia (AD). So far, the findings have been of a quantitative nature. Little is known, however, about the developmental path of the change, its onset, the domains in which it initially occurs, and if and how it spreads to other linguistic domains. The present study investigates pronoun use in six speakers of German a decade before they were clinically diagnosed with AD (LAD) and six biographically matched healthy controls (CTR). The data originate from monologic spoken language elicited by semi-spontaneous biographical interviews. Investigation of nine pronoun types revealed group differences in the use of three pronoun types: D-pronouns—a specific pronoun type of German for reference to persons and objects; the impersonal pronoun *man* ‘one’, and the propositional pronoun *das* ‘this/that’. Investigation of the linguistic contexts in which these three pronoun types were used revealed a correlation with declines in elaborative and evaluative information; that is, information the hearer would benefit from in creating an informed model of the discourse. We, therefore, hypothesize that the early changes in language use due to AD point to problems in perspective-taking, specifically in taking the hearer’s perspective.

## 1. Introduction

A growing number of studies provide evidence that Alzheimer’s Dementia (AD) affects language use long before clinical diagnosis, that is, even prior to clinical diagnosis of mild cognitive impairment. Early deviations from healthy aging controls have been found in, for example, phonemic and semantic verbal fluency [1,2,3], quantity and density of information [4,5], information flow [6], idea density and informativity on referents and locations [7,8,9], lexical richness [4,5], and syntactic complexity [5]. The deviations in these more global features of language use seem to be due to changes in the use of various specific linguistic devices. A reduction in syntactic complexity becomes apparent as reduced use of, for instance, prepositional phrases, function words, conjunctions, and nouns [5,10,11,12,13]. Problems in verbal fluency seem to be related to a lower number of content words (nouns, modifiers) accompanied by more frequent use of pronouns, determiners, and light verbs [4,5,10,11,12,13]. Together, these findings have given rise to the development of computational models predicting the presence or risk of AD on the basis of written or oral text production [6,10,11,14,15,16]. Improving the predictive power of these models and their applicability to different languages is a current and urgent task. Linguistic investigation of relevant domains of language use can not only contribute to identifying and classifying early affected language features but also detect the language-specific manifestations very early pathological changes will have. In addition to improving computational prediction of the risk of AD, finer-grained linguistic results will also help calibrate prevention programs and therapies in the preclinical phase.

The present paper focuses on the use of pronouns 10–12 years before clinical diagnosis of AD. There is robust evidence that pronoun use is one of the early affected linguistic devices in AD. Increased use (For the sake of simplicity, we will speak of “increased” and “decreased” use of pronouns in AD subjects. It is used in the sense of “more frequently/less frequently than in healthy aging controls”. We agree with one reviewer’s objection that in our study, as in most other studies on pronoun use, intra-subjective developments have not been investigated.) is reported for English-speaking AD patients [12,17,18,19,20] and German-speaking AD patients [21,22,23] as well as for Italian-speaking AD patients [13]. Recent studies provide evidence that relevant changes are already present in preclinical and even prodromal stages of AD; see on English [10,15,24] and on German [5,23,25]. Though all these studies found significantly higher proportions of pronouns in the AD groups compared with healthy aging controls, the findings vary in strength and—specifically in the studies on German—in the affected types of pronouns. Some studies have even found no relevant changes [26,27]. The differences mainly result from variation in the method of investigation, most importantly the type of language data and the set of pronouns The type of investigated data is determined by the communicative setting used in data elicitation. Each communicative setting promotes a specific sample of pronouns. For instance, Tönjes [22] used an interactive action setting, which led to a preference for the use of deictic pronouns [20]. Wendelstein investigated monologic spoken language [5], which in German gives rise to a preference for the so-called D-pronouns (*der*, *die*, *den*, etc.) over personal pronouns (*er* ‘he’, *sie* ‘she’, *ihn* ’him’, etc.). The latter are preferred in written texts instead. So far, little attention has been paid to the type of pronouns investigated. Inclusion or exclusion of D-pronouns in German is only one example of variation among the studies. It also makes a big difference whether first- and second-person pronouns (I, you, we, us) are part of the investigated sample. These are of a different qualitative nature than third-person pronouns in that their use is not related to underlying semantic concepts and the appropriate words. Further, there are several pronoun types that have not been taken into consideration so far, for example, possessive pronouns, relative pronouns, interrogative pronouns, indefinite pronouns, and adverbial pronouns. They all provide placeholders for complex referential expressions, but they do not form a homogeneous part of speech. Even pronouns of the same type can have considerably different properties across languages. This is specifically true for personal pronouns, which have been part of the investigated sample in all the studies. For instance, English third-person pronouns (*he*, *she*, *him*, *her*, …) refer to human beings and pets nearly without exception. Reference to inanimate objects (*tree*, *car*, etc.) or abstract concepts (*air*, *love*, etc.) is accomplished with ‘it’. German, in contrast, allows reference to inanimate objects or abstract concepts with the equivalent pronouns *er* ‘he’, *sie* ‘she’, *ihn/ihm* ‘him’, and *ihr* ‘her’. In addition, the D-pronouns mentioned above can also have animate as well as inanimate referents. Sampling types of pronouns together weakens the predictive power of pronoun use in AD diagnosis and makes it hard (if not impossible) to detect the onset and sources of changes in this linguistic domain. Tönjes and Bittner et al. [22,23]—which are to our knowledge the only studies that provide analyses of different types of pronouns—revealed considerable differences in the affectedness of the individual types by AD. Moreover, Bittner et al. found that increased pronoun use is restricted to a very specific referential relation [23]. Pathological changes start small, even in pronoun use. Identifying the early domains requires linguistic structural analyses that take into account the referential properties of the individual pronoun types in and across languages.

The specific nature of the cognitive problems induced by emerging AD and leading to early changes in language use is still largely unknown. The preclinical changes reported in the literature to date seem to indicate multiple extralinguistic sources. For instance, longer pauses and reduced syntactic complexity might be related to declines in attention span and working memory while hampered flow of information and reduced information density, that is, problems in discourse organization, might emerge from problems with executive function (EF). Finally, changes in pronoun use and pronominal reference could result from problems in taking the perspective of the hearer, which is considered an aspect of Theory of Mind (ToM) in linguistics. Problems in ToM have not been in the focus of studies on language use in AD so far. However, according to findings from behavioral, psychological, and neuroimaging studies [28,29], this is certainly possible. In the maximally concise form, EF can be defined as covering abilities that guide goal-directed behavior with updating, shifting, and inhibition as its main parts. ToM was originally coined for the ability to attribute mental states to oneself and to others and to use such attribution to predict and explain behavior. Today, the concept has been broken down into several subcomponents referring to different qualities and sources of mental states. Determining the role and impact of EF vs. ToM capabilities with respect to specific behavioral outcomes is, nevertheless, difficult. Capabilities of both domains are active simultaneously and interact in many situations. Not surprisingly, different studies have reached different conclusions on whether deterioration of EF or ToM is the predominant problem in emerging AD [30,31]; for similar questions on healthy aging see [32,33]. Though disentangling problems with EF vs. ToM in AD is beyond the topic of the present study, our results point to problems in the ability to anticipate the hearer’s perspective on what is being said, that is, in taking someone else’s perspective. This line of interpretation would fit with findings of specific correlations between problems in perspective-taking and types of brain lesions. In a case study of two individuals with different brain lesions, Samson et al. found problems in the inhibition of self-perspective in one individual versus problems in taking someone else’s perspective in the other individual. These were associated with lesions in the right inferior frontal gyrus and the left temporoparietal junction, respectively [34]. These correlations have been confirmed by further studies; see for a summary [30]. Moreover, the group study performed by Le Bouc et al. provided evidence that the correlation between problems in taking someone else’s perspective and lesions in the left temporoparietal junction distinguishes AD from frontotemporal dementia [30]; see also [35].

The present study compares pronoun use in two groups of German-speaking subjects a decade prior to clinical diagnosis of AD in the subjects of the target group while the subjects of the control group remained cognitively healthy. We investigated the complete sample of pronouns produced in semi-spontaneous biographical interviews in order to capture as many changes in pronoun usage as possible. In a previous study, we reported on a significantly higher proportion of D-pronouns—a specific pronoun type of German—in the target group [23]. The aim of the present study was to further determine the pattern of deviations in terms of pronoun types and linguistic contexts to draw conclusions about its possible sources.

## 2. Materials and Methods

The analyses are based on semi-structured biographical interviews conducted orally with 12 participants of the Interdisciplinary Longitudinal Study on Adult Development and Aging (ILSE) [36]. All participants were born between 1930 and 1932 and took part in the baseline (1993–1996) and the follow-up assessments (2005–2007). Each assessment included thorough medical, neuropsychological, and psychiatric examinations and semi-structured interviews, which took 6.0 ± 2.6 h at baseline and 1.8 ± 0.8 h at follow-up on average [37]. Six of the 12 participants had remained cognitively healthy at follow-up (CTR group, *n* = 6) while the other 6 participants were clinically diagnosed as having AD or MCI (*n* = 1). In the following, this latter group is called the LAD group (= Later on AD diagnosed, *n* = 6). The AD was diagnosed according to the NINCDS-ADRDA and the NINDS-AIREN criteria [38] and the MCI was diagnosed according to the criteria of aging-associated cognitive decline [39,40]. In addition to MMSE values, it included further anamnestic, clinical, laboratory–chemical and neuropsychological criteria. All diagnoses were established by two specialists in psychiatry. The CTRs and LADs were carefully matched pairwise for age, gender, years of education, and dialect region (Leipzig or Heidelberg). Table 1 presents the individual characteristics of each subject relevant in the present study. The ILSE study was approved by the local ethical committee of the medical faculty of the University of Heidelberg; written and oral informed consent was obtained from each participant.

The data set is restricted to 12 subjects due to the limited availability of transcribed data that yield carefully matched pairs of a LAD subject and a CTR subject.

All analyses were conducted on the baseline data. That is, all analyzed data were extracted from the transcripts of the semi-structured biographical interviews performed at the baseline assessment of the ILSE study. The interviews were recorded with analogue recording devices on tape or in MP3 format and transcribed manually by student assistants. The transcriptions were checked and corrected by a second transcriber. Participants were invited to freely narrate about their childhood and youth without any time constraints. The interview design elicited predominantly monologic spoken language.

The two basic analyses were performed as proportional analyses of a fixed amount of data. To determine whether there are group differences in the *proportion of nouns and pronouns* we selected the first 700 occurrences of nouns and 3rd-person pronouns from each interview (Section 3.1). Excluded from this analysis were nouns in lexicalized constructions such as *im Grunde* ‘basically’ *(Ich hatte im Grunde nichts dagegen.* ‘I basically had nothing against it.’) and *in der Lage sein* ‘be able’ (*Er war in der Lage, zur Polizei zu gehen*. ’He was able to go to the police.’) as well as immediate repetitions of a noun (phrase) or pronoun. Second, we controlled the *proportion of types of pronouns* by analyzing the first 350 3rd-person pronouns produced by each participant (Section 3.2) (The pronouns selected for the two analyses largely overlap. However, since the analyses do not build on each other, this is not relevant for the individual analyses.) The latter analysis included all types of 3rd-person pronouns that were commonly used in the data. These were nine types in total, see Table 2.

Further analyses were performed on the following types of pronouns: D-pronouns (Section 3.3), the indefinite pronoun *man* ‘one’ (Section 3.5), and the propositional pronoun *das* ‘this/that’ (Section 3.6). Each of these analyses was based on the number of pronouns extracted for the individual type in the *proportion of types of pronouns* analysis.

All pronoun data were manually extracted from the transcripts and sampled and coded in excel files. Coding involved the following features: type of pronoun (see Table 2), case (nom./acc./dat./gen.), number (sg./pl.), prepositional phrase (yes/no), referential relation (clear, potentially clear, unclear), and type of clause (main-/sub-clause). Personal pronouns and D-pronouns were additionally coded for type of referent (family, human, inanimate). Utterances containing more than one pronoun were listed and coded as often as the number of pronouns required (One reviewer expressed concern that patterns of pronoun use may vary due to individual differences in topics or other elements of the interviews and that these limitations significantly diminish the value of the data. We cannot deny individual variation in pronoun use, of course. However, variation in topics is restricted by the fact that all interviews show the same order of questions. With all participants, the main bulk of data stems from reports on their childhood family.).

The existence of D-pronouns in addition to personal pronouns is a specific property of German. Here, we briefly introduce the specifics of this pronoun type. D-pronouns show considerable overlap in nature and use with personal pronouns. In many contexts, the two types can be replaced by each other. However, there are differences in the anaphoric function and the interpersonal emotional connotation of the two types. The differences result from the feature (+distance) carried by the D-pronoun. Anaphorically, D-pronouns are preferably used when referring to discourse entities that are not in the actual center of joint attention. That is, though the referents of D-pronouns have already been introduced in the discourse, they are not the most highly activated (focused) referents. They are at some distance to the center of attention. Using a D-pronoun brings them back into the joint focus of attention. D-pronouns often indicate a topic change while personal pronouns continue the given topic of the discourse. On the interpersonal level, the feature (+distance) introduces a connotation of emotional distance. This is reflected in the fact that using a D-pronoun—instead of the name or a personal pronoun—is considered rude when talking about a person present in the actual situation of discourse. As we will see, however, the (+distance) feature affects pronominal reference to human beings more generally.

For all statistical analyses, chi-square tests were applied using Microsoft Excel (v16.16.27). The significance level was set at 1 percent; therefore, values of *p* < 0.01 were considered significant. Analyses were performed on sum scores calculated in each group for each factor (see Section 2). Boxplots were used to visualize the range of individual variation.

## 3. Results

### 3.1. Proportion of Nouns and Pronouns

Some studies have reported decreases in the use of nouns in the early preclinical and mild stages of AD, see [42]. To find out whether pronoun use is increased at the expanse of nouns in the LAD group, we first controlled the quantitative proportion of nouns and pronouns. Analyses of the first 750 nouns and pronouns produced by each subject revealed no significant group difference in the proportion of both classes (*X^2^*[3] = 0.008, *p* = 0.93). Table 3 presents the results for each individual participant and the total numbers for each group.

Figure 1 illustrates that individual variation is somewhat larger in the LAD group than in the CTR group.

### 3.2. Pronoun Types

The proportional distribution of the nine most commonly produced pronoun types was calculated based on 2100 pronoun uses in each group. This number results from the first 350 pronoun uses of each subject. Figure 2 presents the group comparison for each type of pronoun.

The LAD group significantly differs from the CTR group in the use of *D-pronoun* (D_PRO) and *indefinite pronoun* (INDEF_PRO): *X^2^*[3] = 17.24; *p* < 0.001 and *X^2^*[3] = 10.41; *p* < 0.001, respectively. While the LAD group produced significantly more D-pronouns than the CTR group, it was the opposite case with indefinite pronouns. A marginal effect occurred with *propositional das* (das_propos; *X^2^*[3] = 4.11; *p* < 0.05), which also occurred more often in the CTR group than in the LAD group.

Individual variation, see Figure 3, is especially high in the LAD group’s use of D-pronouns. It is also considerably high in the CTR group’s use of indefinite pronouns.

### 3.3. Summary of the Proportional Analyses

Analysis of the first 750 nouns and pronouns produced by each participant did not reveal a significant group difference in the proportion of nouns and pronouns. Nevertheless, individual variation in the proportion of these two parts of speech is more pronounced in the LAD group. Analysis of the first 350 pronoun productions of each participant revealed significant or nearly significant group differences in the proportion of D-pronouns, indefinite pronouns, and the propositional pronoun *das*. While the proportion of D-pronouns is higher in the LAD group than in the CTR group, the proportion of indefinite pronouns is higher in the CTR group than in the LAD group. The latter is also true for the propositional pronoun *das*, where the group difference almost reaches significance.

In the following section, we ask whether there are specific structures or domains of use that lead to the higher or lower proportion of *D-pronoun*, *indefinite pronoun,* and *propositional das* in the LAD group. Inclusion of *propositional das* in the following analyses is motivated by the finding of a significantly increased group difference at follow-up in a previous study [22]. The marginally significant group difference observed here at baseline might be related to the emerging disease.

### 3.4. D-Pronouns

In a previous study, we found that the increased use of D-pronouns in the AD subjects did not affect structural contexts in which a personal pronoun is preferred over a D-pronoun in German [23]. (Specifically, we investigated the occurrence of D-pronouns and personal pronouns in (i) the postverbal subject position of main clauses and (ii) the subject position of subordinate clauses). The anaphoric function of the two pronoun types seems to be intact until the mild stages of AD. Instead, we found evidence for changes in the referential domain of D-pronouns. This finding is confirmed by the present analyses of a larger sample of baseline data. The analysis is based on the categorization of all referents of singular D-pronouns and singular personal pronouns for one of the following three referential categories: family member (family), non-family human (human), and inanimate referent (inanimate). Figure 4 presents the uses of the two pronoun types in each category in each group.

The subjects of the CTR group tend to prefer personal pronouns over D-pronouns when referring to family members but—vice versa—D-pronouns over personal pronouns when referring to non-family humans. In contrast, the subjects of the LAD group prefer D-pronouns over personal pronouns with both classes of referents. The LAD group’s preference for D-pronouns in the *family* class is significant (*X^2^*[3] = 8.44; *p* < 0.001). In addition, the group difference in the proportion of D-pronouns in the *family* class is significant when controlled against the total number of pronouns (*n* = 2100/group): *X^2^*[3] = 21.22; *p* < 0.001. These findings suggest that the higher proportion of D-pronouns in the LAD group reported in Section 3.2 originates from changes in pronoun use for family members. The potentially pathological nature of this change is underlined by high individual variation in the LAD group’s use of D-pronouns and personal pronouns in this referential class, see Figure 5.

### 3.5. Indefinite Pronouns

The pronoun type *indefinite pronoun* subsumes a broader range of semantically different pronouns, such as *man* ‘one’, *manche* ‘some’, *alle* ‘all’, and *einige* ‘some’. Figure 6 illustrates that the significantly higher proportion of *indefinite pronoun* in the CTR group reported in Section 3.2 is brought about by only one of these pronouns, the impersonal pronoun *man* ‘one’. It is the most frequent indefinite pronoun in both groups. However, the proportion of *man* is significantly higher in the CTR group than in the LAD group when compared with the total number of pronouns (196/2100 and 113/2100, respectively; *X*^2^[3] = 24.06; *p* < 0.001) as well as when compared with the total number of indefinite pronouns produced by each group (196/305 and 113/235, respectively; *X*^2^[3] = 14.19; *p* < 0.001).

Individual variation in the use of *man* is relatively high in both groups; see Figure 7.

As in the analyses of D-pronouns, we asked whether there are specific sources of the group difference in the use of *man*. Analyses revealed a significant difference in the types of finite verbs in the clauses with *man*, specifically in the combination of *man* with an auxiliary construction. This applies to the use of *man* in combination with auxiliary constructions in total, for example, *hat* ‘has’/*hatte* ‘had’/*war* ‘was’ + past participle (102/2100 CTR group and 46/2100 LAD group; *X^2^*[3] = 21.96; *p* < 0.001), as well as to the use of *man* in combination with the most frequent type of auxiliary construction *hat* ‘has’ + past participle (74/2100 CTR group and 34/2100 LAD group; *X^2^*[3] = 15.21; *p* < 0.001) (The group differences in the use of the *man* + auxiliary construction in general and of *man* + *hat* + past participle are significant even when compared to the total number of indefinite pronouns produced by each group (305 CTR group and 235 LAD group): *X^2^*[3] = 12.83; *p* < 0.001 and *X^2^*[3] = 7.96; *p* < 0.001, respectively). Figure 8 illustrates the results of both analyses.

### 3.6. Propositional das

The group difference in the proportion of the propositional pronoun *das* ‘this/that’ is only marginally significant when compared with the total number of pronouns (Section 3.2). Nevertheless, we also performed specific analyses of the use of this pronoun type. As said in Section 3.3, this is motivated by the finding of a significant group difference in the proportion of this pronoun type at follow-up [23]. Again, we performed an analysis of the type of finite verb occurring in the clauses with the propositional pronoun *das*. Figure 9 shows that the most frequent type of finite verb is a copula, such as *ist* ‘is’, *sind* ‘are’, *war* ‘was’, and *waren* ‘were’. The group difference in the proportion of copula verbs is marginally significant (*X^2^*[3] = 6.04; *p* = 0.01) when compared with the total number of pronouns (236/2100 CTR group and 188/2100 LAD group).

In both groups, the most frequent copula is the past tense copula *war*/*waren* ‘was/were’ (190 CTR group and 145 LAD group). The group difference in the proportion of *das* + past tense copula is also marginally significant when compared with the total number of pronouns (*X^2^*[3] = 6.57; *p* = 0.01). Comparison of the use of present and past copulas in clauses with the propositional pronoun *das* illustrates that it is indeed the past construction that brings about the group difference (Figure 10).

## 4. Discussion

The present study investigated whether there is evidence for pathological changes in pronoun use of German speakers a decade prior to the clinical diagnosis of AD, and, if so, in which domains of pronoun use these early changes occur. The study compared pronoun use in six speakers who would be diagnosed with AD a decade after the elicitation of the investigated data (LAD group) and six speakers who have remained cognitively healthy after this decade (CTR group).

By conducting proportional analyses, we found significant group differences in the use of three types of pronouns. These differences do not result from replacement of nouns by pronouns in the LAD group. Crucially, we not only observed increases in pronoun use in the LAD group but also decreases. The increase found with D-pronouns is in line with findings reported in the literature. Decreases, however, have not been reported so far (The only exceptions we are aware of are Gress-Heister, who reports a decreased use of personal pronouns in five persons with different cognitive impairments [26], and Tönjes, who reports a significantly lower percentage of anaphorically used D-pronouns in his AD group [22]. In the latter case, this is a side effect of the dominance of deictic D-pronouns in the AD data.). In our data, they occurred with the indefinite pronoun *man* ‘one’ and the propositional pronoun *das* ‘this/that’. Analyses of the linguistic contexts in which these three pronoun types are used supported our assumption that problems in perspective-taking reported in the literature on AD are a likely source of the observed pathological changes. Specifically, our results point to problems in inferring the hearer’s needs in establishing a coherent mental model of what s/he has been told by the speaker. In the following, we will support this conclusion by discussing our key findings.

### 4.1. Increased Use of D-Pronouns

The finding of a significantly higher proportion of D-pronouns in the LAD group compared with the CTR group is in line with previous findings on pronoun use in German-speaking AD and LAD subjects [5,21,22,23,25]. However, across these studies, there is considerable variation in the type of pronoun found to be increased in use, which results from variation in the set of pronouns and the type of language data investigated. Analyses of individual pronoun types are provided only by Tönjes and Bittner et al. [22,23]. Though both studies found a significant increase in the use of D-pronouns, the increase occurs with different variants of the pronoun type. Tönjes analyzed dialogic data elicited during joint manipulative interaction, namely baking a cake together. This interactive situation promotes deictic use of pronouns (e.g., *Please*, *give it*/*this*/*that to me.*). Thus, the most prominent increase in pronoun use appeared with deictic D-pronouns in this study. Bittner et al.—as well as the present study—only investigated monologic spoken language concerning topics not bound to the actual situation of communication. Deictic use of pronouns is not prominent in this communicative setting. However, spoken German favors the use of D-pronouns over personal pronouns [43] (In written German, it is typically the other way around. That is, personal pronouns are preferred over D-pronouns. Therefore, studies based on written German are more likely to find increased use of personal pronouns in AD data. The type of data investigated, obviously, has a great impact on the set of pronouns applied and, thus, on where, i.e., in which linguistic domains, pathological changes come to light.). This explains why the CTR group also produced a higher proportion of D-pronouns than of personal pronouns and why we found increased use of (non-deictic) D-pronouns but not of personal pronouns in the LAD group. Linguistically, the prominence of D-pronouns in spoken German is caused by their specific capacity to increase the level of attention on referents, which is specifically useful in situations with a high load of non-linguistic contextual information [44]. In sum, it is reasonable and even expectable that an increase in pronoun use due to AD affects D-pronouns in spoken German.

The analysis of the linguistic contexts in which the D-pronouns are used showed that the increase is not spread across all contexts of D-pronoun use. It appears to be restricted to references to family members. While the CTRs preferred personal pronouns over D-pronouns for these emotionally close referents, the LADs preferred D-pronouns over personal pronouns (Figure 4). In addition, the LADs used a somewhat higher proportion of personal pronouns than the CTRs when referring to non-family humans, that is, when referring to emotionally less close referents. Together, the two findings suggest problems with the connotation of emotional distance carried by the D-pronoun, which is an application of the feature (+distance) carried by D-pronouns in general (see Section 2 Applying a D-pronoun to a person—when it is not required for attention-guiding reasons—assigns an emotional distance of the speaker to the referent or his/her actions or behavior. Inaccuracies in stating or not stating this information result in the hearer being insufficiently or even wrongly informed about the emotional attitude of the speaker towards the person referred to or their actions. This makes it difficult for the hearer to create a fully adequate mental model of the discourse (Pronoun use is, of course, only one linguistic means for signaling emotional affectedness. However, using a D-pronoun when a personal pronoun would be expected is an explicit expression of emotional distance).

Before drawing initial conclusions, it is worth noticing that the observed use of D-pronouns in references to emotionally close people may be encouraged by the context of the data analyzed here. The biographical interviews conducted in the ILSE study motivate participants to talk about aspects of their personal lives in childhood and family situations. This naturally leads to higher proportions of pronouns referring to emotionally close people. Nevertheless, it has to be taken into account that disturbances with the emotional connotation of D-pronouns may correspond to changes in the emotional constitution of the LAD subjects (Correlation of later AD with reduced emotional expression is also reported from the precursor study [7,8]). Given the complete pattern of pronoun use found in the LAD group we would, however, emphasize another aspect of the finding. Confusion in the emotional connotation of pronoun use makes it difficult for the hearer to create a fully informed mental model of the discourse, that is, a model that also contains metacommunicative information on the speaker’s stances or attitudes toward the factual information. On the speaker’s side, this suggests problems in inferring the communicative needs and, possibly, the mental states of the hearer. In the following, we argue that the observed decreases in the use of the indefinite pronoun *man* ‘one’ and the propositional pronoun *das* ‘this/that’ provide further evidence that the early pathological changes in language use are caused by problems in taking the hearer’s perspective and satisfying the hearer’s needs in creating a fully informed mental model of the discourse.

### 4.2. Decreased Use of the Pronouns man ‘one’ and das ‘this/that’

At first glance, it seems quite natural that a proportional analysis of a fixed set of data will show decreases in some parts of the data set when other parts show increases. Concerning our data set of nine pronoun types, the larger proportion of D-pronouns in the LAD group than in the CTR group could have led to decreased proportions of all other types of pronouns or at least most of them. However, this is not the case. Decreased pronoun use occurs with only two specific pronouns, the indefinite pronoun *man* ‘one’ and the propositional pronoun *das* ‘this/that’. With each of these two pronouns, we found that the decrease was restricted to specific domains of use. With *man,* the significant group difference was restricted to clauses containing *man* and an auxiliary + past participle, most frequently *man* + auxiliary *hat* ‘has’ + past participle. With the propositional pronoun *das,* the significant group difference was restricted to clauses containing *das* in subject function and the past tense copula *war/waren* ‘was/were’.

Both the clauses with *man + hat + past participle* and the clauses with subject *das + war/waren* are linguistic structures typically used for information that elaborates or evaluates factual information conveyed previously. They provide background knowledge that supports appropriate integration of the factual information and of the speaker’s attitudes about it. The latter is especially obvious in the examples given in (2c). It is, however, also present in many of the *man + hat + past participle* constructions. Here, the evaluative attitude of the speaker is often underlined by specific discourse particles called *Abtönungspartikel*, such as *mal*, *eben*, *ja*, *auch*, and *aber* (there is no English translation of these particles that captures the meaning they have when used as *Abtönungspartikel*). To illustrate the evaluative and elaborative nature typical of these constructions, we present some examples from the data.

(1)
*man + hat + past participle*
*viel spielsachen hat man ja nicht gehabt.*                    (CTR2)‘one did not have many toys.’*man hat sich auch mal bisschen gezankt.*                    (CTR4)‘one also quarreled a bit.’*aber aber da hat man als kind schon mitgekriegt dass der krieg im anflug war.*    (CTR5)‘but one noticed as a child that the war was on its way.’*und man hat viel viel spaß gehabt auch.*                    (LAD1)‘and one had a lot of fun too.’*da hat man ja noch kein auto gehabt.*                     (LAD3)‘then one did not have a car yet.’*wenn es nicht geklappt hat, da hat man eben mal eine hinter die ohren gekriegt.*  (LAD5)‘if it did not work out, one just got one behind the ears.’
(2)subject *das + war/waren*
elaboration on location and time
*das war das hinterhaus.*                          (CTR1)‘this was the rear house.’*ja das war die schulzeit.*                          (CTR6)‘yes, this was the school time.’*das war dann in der heilstätte.*                       (LAD6)‘that was in the sanatorium then.’*das war ach da war so ein kalter winter.*                   (LAD4)‘that was oh there was such a cold winter.’elaboration on objects and persons
*also das war eine junge großmutter.*                     (CTR4)‘so that was a young grandmother.’*das waren die alten lehrer.*                         (CTR5)‘these were the old teachers.’*ja aber ansonsten waren das nur immer abgetragene kleidungen von den älteren.* (LAD6)‘yes but otherwise these were just worn-out clothes from the older ones.’*und das war ein arzt der war früher auch bei der armee.*             (LAD5)‘this was a doctor who was in the army before too.’speaker judgments/attitudes
*dass er da arg deprimiert war, weil das bloß so vorübergehend die arbeit war.*  (CTR2)‘that he was very depressed, because that was just temporary, the work(ing place).’*das war damals ja ganz normal.*                       (CTR6)‘this was completely normal at that time.’*meine mutter war hausfrau, wie das früher war.*                (CTR7)‘my mother was a wife as was normal at that time.’*das war mein glück sonst wäre ich auch mit hochgegangen.*          (LAD1)’that was my luck otherwise I would have gone up too.’*das war das war sehr sehr sehr hart diese zeit.*                 (LAD2)’that was that was really really really hard those days.’*das war eine seele von mensch.*                       (LAD4)’this (=she) was a good soul.’

### 4.3. Individual Variation in the LAD Group

As is to be expected, there is considerable variation in the strength of the deviation of the individual LAD subjects from the CTRs. To some extent, it is a matter of individual styles of communication, which is also the case, for instance, with the variation in the use of *man* ‘one’ within the CTR group (Figure 7).

Leaving this aside, AD manifests itself in slightly different ways in each person, and so the concrete patterns of initial changes in language use will vary among the individual subjects. Two aspects of our findings on individual variations, however, suggest that the observed deviations in pronoun use are caused by the emerging AD. First, we found greater individual variation in the LAD subjects than in the CTR subjects in nearly all analyses and, second, the modest deviation from the CTRs typically appeared with the only subject of the LAD group that had a MCI diagnosis at follow-up instead of an AD diagnosis (see LAD6 in Table 1). Given that this assumption is correct, the latter finding would suggest, in addition, that the changes we observed in the AD group start about a decade before AD can be clinically diagnosed by the current diagnostics. The alternative possibility would be that the later AD subjects performed a different pattern of pronoun use from their youth on.

In general, we are aware, of course, that the data base of our study is very small, which limits the robustness of our results and conclusions. Further studies are needed to determine whether the findings would be borne out in a larger number of subjects and what conclusions should be drawn regarding individual patterns.

### 4.4. Problems in Perspective-Taking as a Source of Preclinical Pathological Changes in Pronoun Use

Contrary to what was proposed earlier on, the initial changes in pronoun use are not caused by semantic deficits that hamper the activation of nouns. Instead, our results suggest that the pathological changes start in linguistic contexts typically used to provide elaborative and evaluative information on what is being communicated. This information is additional to the factual information conveyed. It has the function of supporting the hearer in correctly understanding what the speaker wants to tell, that is, in establishing the desired mental model of discourse. This includes not only appropriate understanding of the factual information but also, and especially, of the speaker’s attitudes about it. To achieve this goal, a speaker must consistently update the hearer’s state of factual and evaluative knowledge as well as his perspective on it. Obviously, healthy aging people are more active in giving additional, non-factual information than people with emerging AD. This suggests that the former are more active in taking the hearer’s perspective than the latter. In the present analysis, the LAD subjects’ problem in providing elaborative and evaluative information is clearly documented by the less frequent use of the constructions *man + hat + past participle* and *subject das + war/waren*. In our view, the increased use of D-pronouns in references to emotionally close persons results from the same underlying problem. It appears to be a reduction in the evaluative information on the speaker’s attitude towards the person referred to. As we have shown in Bittner et al., there are no differences in the LADs’ and the CTRs’ use of D-pronouns and personal pronouns on the structural level [23]. That is, the anaphoric function of both pronoun types in ensuring discourse coherence is not affected in the investigated stage. The only affected aspect is the evaluative information.

Our results fit with behavioral and neuroimaging studies that have found problems in taking the perspective of others specific to AD. In these studies, the problem manifested itself in extra-linguistic tasks. Samson et al. and Le Bouc et al. applied different versions of false belief tasks [30,34]. Dermody et al. used the Interpersonal Reactivity Index of Davis [35,45]. Crucially, these studies also showed that the problem in taking the perspective of others correlates with neurological abnormalities in the left temporoparietal junction, which distinguishes AD from other forms of dementia, specifically frontotemporal dementia; see also [29].

There are differences in assigning the perspective-taking problem of AD patients to problems in the domain of ToM versus the domain of EF or even to other cognitive domains; for the latter, see [46]. While Samson et al. did not touch on the issue [34], Le Bouc et al. and Dermody et al. consider ToM as the relevant domain [30,35]. Others consider the ability to shift perspective a tool of behavior regulation that is part of EF [28,31]. From a linguistic point of view, perspective-taking in the sense of anticipating someone else’s perspective on what is communicated is a component of mind-reading, that is, of ToM. However, mind-reading abilities can be hampered by problems in other cognitive domains. Our data do not allow further conclusions on the final sources of the LADs’ problems in taking the hearer’s perspective. In our view, however, they exclude attention deficits [1} and declines in working memory [17] as being the primary sources of the observed changes. If these were behind the difficulties, the patterns of change should be much more heterogeneous rather than being restricted to elaborative and evaluative information. Declines in these domains are more likely to be primary sources of reductions in syntactic complexity than of problems in providing elaborative and evaluative information.

Though our study was restricted to pronoun use, the results suggest that the underlying problem of anticipating the hearer’s perspective affects language use beyond the use of pronouns. Other potentially affected language domains are, for instance, the *Abtönungs-partikel* mentioned above and the epistemic use of modal verbs. Both add connotations of speaker attitudes such as certainty or uncertainty, irony, or expectedness vs. unexpectedness.

## 5. Conclusions

Our study provides evidence that changes in pronoun use start a decade before clinical diagnosis of AD. Analyses of the linguistic contexts of the changes have revealed a restriction to elaborative and evaluative information. We, therefore, hypothesize that early changes in pronoun use emerge from a more global problem in language production, which is taking the perspective of the hearer. Our results fit with findings of behavioral and neuroimaging studies showing that taking someone else’s perspective is specifically problematic in AD and correlates with brain lesions in the left temporoparietal junction.

Our study also provides evidence that the concrete linguistic structures that are affected first by AD are widely language-specific in nature. D-pronouns and the emotional connotation they have in reference to humans as well as the affected type of use of the impersonal pronoun *man* ‘one’ are clearly language-specific properties of German. Other languages will show other patterns of pathological change in the preclinical stages. We would, however, hypothesize that the initial changes in language use that are caused by an emerging AD affect elaborative and evaluative information in all languages.

## Figures and Tables

**Figure 1 brainsci-12-00121-f001:**
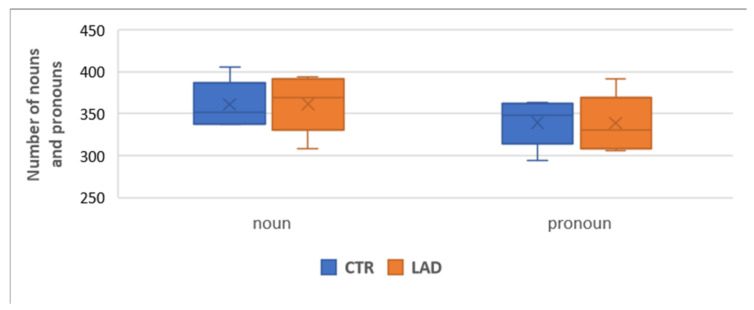
Range of proportions of nouns and pronouns in the CTR group and the LAD group.

**Figure 2 brainsci-12-00121-f002:**
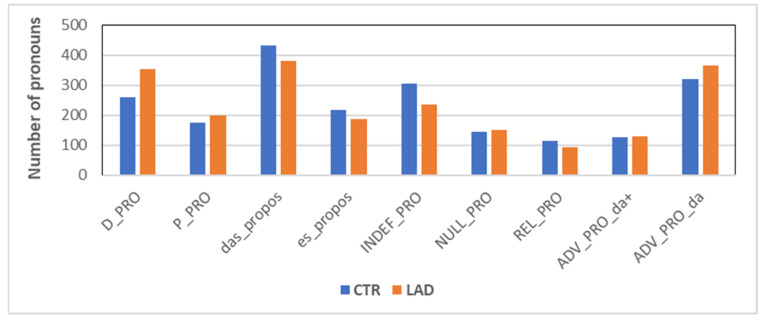
Proportion of pronoun types (for abbreviations and examples, see Table 2).

**Figure 3 brainsci-12-00121-f003:**
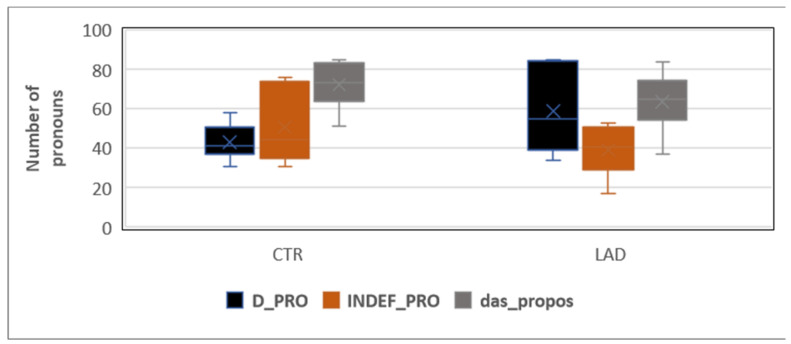
Range of proportions of *D-pronoun* (D_PRO), *indefinite pronoun* (INDEF_PRO), and *propositional das* (das_propos) in the CTR group and the LAD group.

**Figure 4 brainsci-12-00121-f004:**
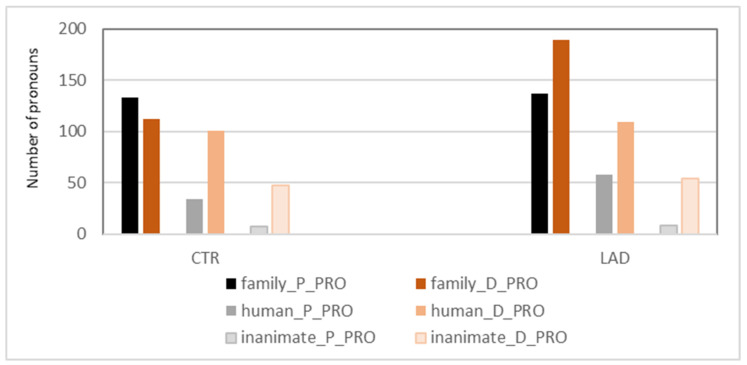
Classes of referents of personal pronouns and D-pronouns (singular pronouns only).

**Figure 5 brainsci-12-00121-f005:**
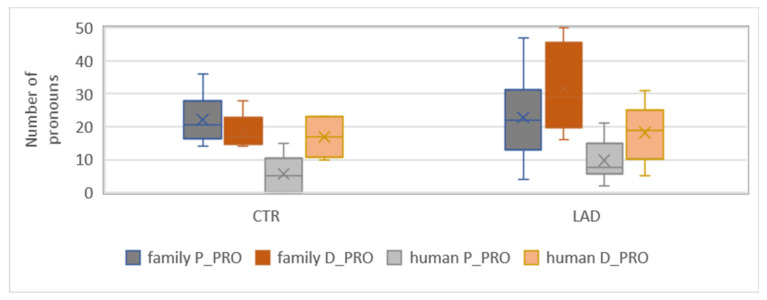
Range of proportions of *D-pronoun* (D_PRO) and *personal pronoun* (P_PRO) in references to family members and (other) humans in the CTR group and the LAD group.

**Figure 6 brainsci-12-00121-f006:**
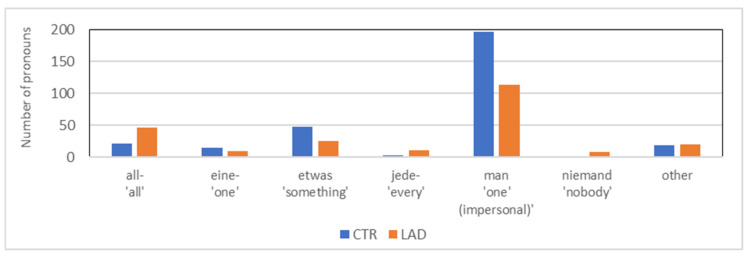
Types of indefinite pronouns in the CTR group and the LAD group.

**Figure 7 brainsci-12-00121-f007:**
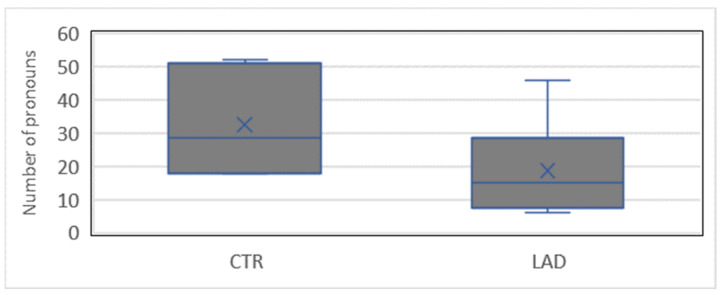
Range of proportions of impersonal *man* ‘one’ in the CTR group and the LAD group.

**Figure 8 brainsci-12-00121-f008:**
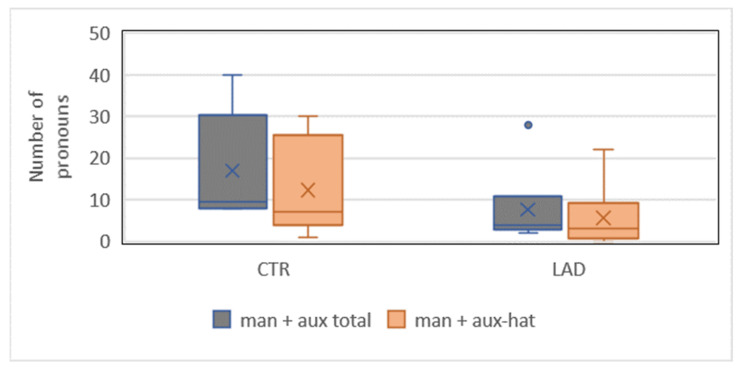
Proportion of *man* ‘one’ + auxiliary in total and *man* + auxiliary *hat* ‘has’ in the CTR group and the LAD group.

**Figure 9 brainsci-12-00121-f009:**
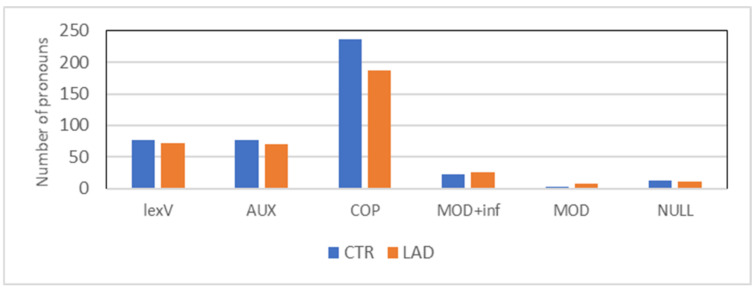
Types of finite verbs in clauses with the propositional pronoun *das* ‘this/that’.

**Figure 10 brainsci-12-00121-f010:**
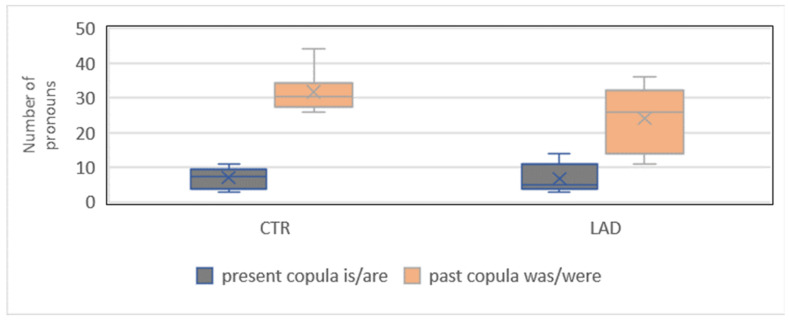
Proportion of propositional pronoun *das* + present tense copula and + past tense copula.

**Table 1 brainsci-12-00121-t001:** Sample description.

Subject ID	Age (Baseline)	Age (Follow-Up)	Sex	Region	Education in Years	Diagnosis (Baseline)	Diagnosis (Follow-Up)	MMSE (Follow-Up)
CTR1	63	73	m	HD	13	CI	CI	30
CTR2	62	73	m	HD	12	CI	CI	29
CTR3	62	73	f	HD	9	CI	CI	26
CTR4	64	75	m	LE	13	CI	CI	27
CTR5	63	74	m	LE	18	CI	CI	30
CTR6	64	74	m	HD	12	CI	CI	29
LAD1	62	74	m	HD	13	CI	AD	24
LAD2	63	74	m	HD	12	CI	AD	25
LAD3	62	74	f	HD	9	CI	AD	25
LAD4	64	75	m	LE	13	CI	AD	25
LAD5	64	74	m	LE	14	CI	AD	26
LAD6	63	74	m	HD	12	CI	MCI	26

Note: CI, cognitively intact; AD, Alzheimer’s Dementia; HD, Heidelberg (Germany); LE, Leipzig (Germany); MCI, mild cognitive impairment; MMSE, Mini Mental State Examination to examine the severity of cognitive deficits [41]. The MMSE was not applied at baseline; therefore, data were not available until follow-up.

**Table 2 brainsci-12-00121-t002:** Types of pronouns included in the proportional analysis of types of pronouns.

Type of Pronoun	German Forms	English Equivalents
Personal pronoun (P_PRO)	*er*, *sie*, *es*, *ihn*, *ihm*, *ihr*, *sie*, *ihnen*, *ihren*, *…*	*he*, *she*, *it*, *him*, *her*, *they*, *them*, *their*, *…*
D-pronoun (D_PRO) + demonstrative pronoun *dies*- ^1^	*die*, *der*, *das*, *den*, *dem*, *denen* *diese*, *dieser*, *diesen*, *…*	*–* *this*, *that*
NULL-pronoun (NULL_PRO)	coordinated clauses: Sie schlafen und *Ø* träumen. ellipses: *Ø* wollte noch kommen	They are sleeping and *Ø* are dreaming. *Ø* wanted to come.
Propositional *das*(das_propos)	Du bist traurig, ich weiss *das*.	You are sad, I know *it*/*that*.
Propositional *es*(es_propos)	Du bist traurig, ich weiss *es*.	You are sad, I know *it*.
Indefinite pronoun (INDEF_PRO)	*manch-*, *einige*, *man*, *ein-*	*some*, *someone*, *one*, *one of*
Relative pronoun (REL_PRO)	der Hund, *der* nachts bellt	the dog *that* barks at night
Adverbial pronoun *da*+ (ADV_PRO_da+)	*darum*, *dafür*, *damit*, *…*	*therefore*, *for this*, *in order to*, *…*
Adverbial pronoun *da* (ADV_PRO_da)	am Sonntag, *da* war sie tanzen in Berlin, *da* tanzt sie. es war Musik, *da* haben sie getanzt.	on Sunday, (on that day) she was dancing. in Berlin, (in that town) she dances. there was music, *so* they were dancing.

^1^ Demonstrative pronouns (*diese*, *dieser*, etc.) rarely occurred in the data (*n* = 12). All occurrences were included in the class of D-pronouns since they showed the same referential relations.

**Table 3 brainsci-12-00121-t003:** Proportion of nouns and pronouns (database: first 750 nouns and pronouns produced by each subject).

	Noun	Pronoun			Noun	Pronoun
CTR-1	337	363		LAD-1	308	392
CTR-2	338	362		LAD-2	391	309
CTR-3	380	320		LAD-3	385	315
CTR-4	363	337		LAD-4	338	362
CTR-5	341	359		LAD-5	394	306
CTR-6	406	294		LAD-6	353	347
CTR-total	2165	2035		LAD-total	2169	2031

## Data Availability

Data are available from the authors on request (J.S. or C.F.).

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
