# Peer review of "Changes in Pronoun Use a Decade before Clinical Diagnosis of Alzheimer’s Dementia—Linguistic Contexts Suggest Problems in Perspective-Taking"

_brainsci, 2022, doi:10.3390/brainsci12010121_

Round 1

Reviewer 1 Report

The present paper focuses on the analysis of linguistic features -specifically, pronouns- as preclinical language-related marker of Alzheimer’s disease (AD). Currently, studies based on such an approach are more than welcome in the research on language traits of dementia. AD is a clinical picture with important speech and language-related symptomatology, which is, unfortunately, understudied in the present-day literature. On those grounds alone, the present paper would be more than welcome for publication.

Yet, there are other several strong points I believe to make the present paper eligible (and very welcome) for publication. The application of linguistic perspective to the study of discourse in AD is appropriate and correct; as a linguist, I am extremely happy to read a language-based research of AD symptomatology. The methodology itself is robust and follows a strong hypothesis merging a language-based and a neuropsychology-based assumptions. Indeed, it is a truly interdisciplinary research I have been very happy to assess.

In what follows, I simply comment on several aspects of the study I would like authors to consider for possible inclusion in their final version of the manuscript.

  • Lines 96-99: When saying “longer pauses and reduced syntactic complexity might be related to declines in attention span and working memory while hampered flow of information and reduced information density, that is, problems in discourse organization, might emerge from problems with executive functions (EF)”, I would suggest mentioning also the disruption of lexical access and, further, of semantic networks, which are highly responsible for primary anomia and then, for lexical-semantic deficit. The same observation could work for what is mentioned in lines 100-102, when the authors state “Finally, changes in pronoun use and pronominal reference could result from problems in taking the perspective of the hearer, which is considered an aspect of Theory of Mind (ToM) in linguistics”: increase in the use of pronouns could also be a consequence of anomia, which could act as fillers or compensatory strategies (just like formulaic units, for example). Although at the end of manuscript authors state that changes in the use of pronouns do not fill in referential gaps, in some other languages (for example, Spanish, which I work with) pronouns do boost as a result of discourse compensation.
  • Lines 114-117: "Disentangling problems with EF vs. ToM in AD is beyond the topic of the present study. However, we hypothesize that the seemingly different changes in pronoun use that we observe in the AD-group result from a decline in the ability to anticipate the hearer’s perspective on what is being said, that is, in taking someone else’s perspective” > It is not clear to me here why the increase in the use of pronouns could be due to the disruption in the ToM. Just as a concern, ToM is not typically affected by AD, but rather by other dementias (bFTD, SD, LBD, etc.). Although some lines below the authors support this observation by results from neuroimaging studies, other types of cognitive and self-perception disorders usually appear first in AD. Second, I do not see a direct or logical relation between the disruption of the ToM and the use of pronouns, at least at this point of the paper. This information seems a bit misleading to me in the context of the present research. I would encourage authors to provide more detailed argumentation of this aspect in order to make it clearer to the readers.
  • In what deals with the target group of subjects (LAD), I am concerned with the inclusion of the patient with MCI (n=1) into the group of AD diagnosed patients. MCI does not always evolve into AD (it is only about approximately 20% of MCI cases that evolve into AD), so how sure can we be about the homogeneity of this subsample? I am also concerned about MMSE scores: they are highly comparable across groups and for the LAD group there is only one person with MMSE=24, which is considered as a baseline for normal cognition even in highly educated speakers. Maybe, the authors should provide information on which criteria specifically were used for discriminating AD from CI speakers.
  • Concerning proportional analysis, why did authors limit the studied material to 750 and 350 first units? Did these units overlap among them? (750 units corresponded to nouns and pronouns and 350, to pronouns only; were these the same pronouns)? How 750 units (the first analysis) were distributed among nouns and pronouns? I lack all this information to make clear what exactly was analyzed in the interview sample (for example, in line 218 the statement is misleading: “Analyses of the first 750 nouns and pronouns produced by each subject”). Maybe, a simple clarification could help at this point.
  • Lines 248-249: “Nevertheless, individual variation in the proportion of these two parts of speech is more pronounced in the LAD-group”. I would encourage authors to develop a little bit more on this observation: in what exactly such individual variation consists?
  • Line 263: “In a previous study found that the increased use of D-pronouns…” > there must be a subject missing (“We/they” found…”)
  • Lines 336-337: “The present study investigated whether there is evidence for pathological changes in pronoun use of German speakers a decade prior to the clinical diagnosis of AD” > I am a bit concerned about the appropriateness of use of the term “pathological changes” when referring to the use of pronouns in speakers with AD. I would rather speak about AD-related, but not pathological use. Indeed, the main question in this respect is how we can explain the increase of D-pronouns against other pronouns in preclinical dementia.
  • Lines 379-381: “While the CTRs preferred personal pronouns over D-pronouns for these emotionally close referents, the LADs preferred D-pronouns over personal pronouns (Figure 4).” > Maybe, considering emotional re-arrangement in AD can also explain this observation. This particular idea seems more logical for me than the ToM itself (or their combination could work better).
  • Line 474, section 4.3. I do understand authors’ position on that variability across LAD speakers could support the idea that at the moment of the baseline interview dementia was already in progress, but this is very difficult to corroborate in an absolute way. To which extent can we be sure, without neuromaging or blood-test support, that 10 years before ongoing AD speakers were already developing dementia? I do not contradict the position of the authors, but rather suggest to be more cautious in this respect (consider including this aspect in the paper itself).

Author Response

Revisions due to comments of Review 1

  • Lines 96-99: When saying “longer pauses and reduced syntactic complexity might be related to declines in attention span and working memory while hampered flow of information and reduced information density, that is, problems in discourse organization, might emerge from problems with executive functions (EF)”, I would suggest mentioning also the disruption of lexical access and, further, of semantic networks, which are highly responsible for primary anomia and then, for lexical-semantic deficit. The same observation could work for what is mentioned in lines 100-102, when the authors state “Finally, changes in pronoun use and pronominal reference could result from problems in taking the perspective of the hearer, which is considered an aspect of Theory of Mind (ToM) in linguistics”: increase in the use of pronouns could also be a consequence of anomia, which could act as fillers or compensatory strategies (just like formulaic units, for example). Although at the end of manuscript authors state that changes in the use of pronouns do not fill in referential gaps, in some other languages (for example, Spanish, which I work with) pronouns do boost as a result of discourse compensation.

We agree with the reviewer that hampered lexical access and interruption of semantic networks is a core problem in AD and that pronouns boost as a result of these processes in the clinical stages. In the relevant paragraph we aimed at putting the focus of extralinguistic sources causing the initial changes in language use. We made this more explicit by inserting “extralinguistic” in line 95: “seem to indicate multiple extralinguistic sources”. Further, we tried to strengthen the focus on the preclinical stages by rewriting the very first §.

  • Lines 114-117: "Disentangling problems with EF vs. ToM in AD is beyond the topic of the present study. However, we hypothesize that the seemingly different changes in pronoun use that we observe in the AD-group result from a decline in the ability to anticipate the hearer’s perspective on what is being said, that is, in taking someone else’s perspective” > It is not clear to me here why the increase in the use of pronouns could be due to the disruption in the ToM. Just as a concern, ToM is not typically affected by AD, but rather by other dementias (bFTD, SD, LBD, etc.). Although some lines below the authors support this observation by results from neuroimaging studies, other types of cognitive and self-perception disorders usually appear first in AD. Second, I do not see a direct or logical relation between the disruption of the ToM and the use of pronouns, at least at this point of the paper. This information seems a bit misleading to me in the context of the present research. I would encourage authors to provide more detailed argumentation of this aspect in order to make it clearer to the readers.

Following the reviewers suggestion we inserted: “Problems in ToM have not been in the focus of studies on language use in AD so far. However, according to findings from behavioral, psychological, and neuroimaging studies [29,30], this is certainly possible.” Line 101-104

  • In what deals with the target group of subjects (LAD), I am concerned with the inclusion of the patient with MCI (n=1) into the group of AD diagnosed patients. MCI does not always evolve into AD (it is only about approximately 20% of MCI cases that evolve into AD), so how sure can we be about the homogeneity of this subsample?

The database is quite small due to the fact that in the available pool of transcribed data the number of LAD/CTR-participants that give a perfectly matched pair is low. We decided to include the MCI-subject and by that the 6th matched pair of subjects since the MCI in this subject persisted in the 4th wave of the study, suggesting that it finally leads to AD.

  • I am also concerned about MMSE scores: they are highly comparable across groups and for the LAD group there is only one person with MMSE=24, which is considered as a baseline for normal cognition even in highly educated speakers. Maybe, the authors should provide information on which criteria specifically were used for discriminating AD from CI speakers.

The criteria listed in the text present the diagnostic criteria for AD applied in the 1990ies. In addition to MMSE, diagnosis of early AD includes further criteria. To make this explicit we added line 149-151: In addition to MMSE-values this included further anamnestic, clinical, laboratory-chemical and neuropsychological criteria. All diagnoses were established by two specialists in psychiatry.

  • Concerning proportional analysis, why did authors limit the studied material to 750 and 350 first units? Did these units overlap among them? (750 units corresponded to nouns and pronouns and 350, to pronouns only; were these the same pronouns)? How 750 units (the first analysis) were distributed among nouns and pronouns? I lack all this information to make clear what exactly was analyzed in the interview sample (for example, in line 218 the statement is misleading: “Analyses of the first 750 nouns and pronouns produced by each subject”). Maybe, a simple clarification could help at this point.

We tried to clarify this point by the following reformulation, line 172-173: To determine whether there are group differences in the proportion of nouns and pronouns we selected the first 700 occurrences of nouns and pronouns from each interview. AND by footnote 3: The pronouns selected for the two analyses largely overlap. However, since the analyses do not build on each other, this is not relevant for the individual analyses

  • Lines 248-249: “Nevertheless, individual variation in the proportion of these two parts of speech is more pronounced in the LAD-group”. I would encourage authors to develop a little bit more on this observation: in what exactly such individual variation consists?

We agree that this is an interesting topic. However, in the current study, the focus is on changes in pronoun use. This first analysis aimed at excluding beforehand that pronoun use expands in the LAD-group merely as a result of a general reduction in the use of nouns as it was found for the clinical stages of AD.

  • Line 263: “In a previous study found that the increased use of D-pronouns…” > there must be a subject missing (“We/they” found…”)

We corrected this.

  • Lines 336-337: “The present study investigated whether there is evidence for pathological changes in pronoun use of German speakers a decade prior to the clinical diagnosis of AD” > I am a bit concerned about the appropriateness of use of the term “pathological changes” when referring to the use of pronouns in speakers with AD. I would rather speak about AD-related, but not pathological use. Indeed, the main question in this respect is how we can explain the increase of D-pronouns against other pronouns in preclinical dementia.

To our view, if a change is AD-related it is a pathological change in the sense that it is caused by the disease and does not occur in healthy aging people. The term, thus, distinguishes changes in language use that are due to AD from those occurring in healthy aging people.

  • Lines 379-381: “While the CTRs preferred personal pronouns over D-pronouns for these emotionally close referents, the LADs preferred D-pronouns over personal pronouns (Figure 4).” > Maybe, considering emotional re-arrangement in AD can also explain this observation. This particular idea seems more logical for me than the ToM itself (or their combination could work better).

We accordingly rewrote the $ Line 398-415 and inserted footnote 11. We agree that the change in D-pronoun use might be supported by the fact that it is an emotional connotation that seems to become ignored. However, assuming emotional problems as the primary background of this finding, would require assuming different sources for changes in the use of D-pronouns vs. impersonal man and propositional das. Having shown that there might be a common source for the three changes is – to our view – the most important contribution of the study to the actual research on language use in the prefield of AD. (see also last point of review 2: “…, I find the value of the paper to be the hypotheses it raises around perspective-taking as indexed by the pronoun use in elaborative and evaluative contexts of the discourse. This seems well beyond the scope of what this study can address but it suggests a cross-linguistic study might be designed to test it out … .”)

  • Line 474, section 4.3. I do understand authors’ position on that variability across LAD speakers could support the idea that at the moment of the baseline interview dementia was already in progress, but this is very difficult to corroborate in an absolute way. To which extent can we be sure, without neuromaging or blood-test support, that 10 years before ongoing AD speakers were already developing dementia? I do not contradict the position of the authors, but rather suggest to be more cautious in this respect (consider including this aspect in the paper itself).

We absolutely agree! We revised the § accordingly.

Reviewer 2 Report

This is a thought-provoking manuscript despite the work's significant limitations. First, the sample size is very small, and the results of the many comparisons must be viewed with caution. Second, the speech samples are taken at only one point in time, prohibiting any speculation of the evolution of pronoun pattern use in cognitive decline (note that the use of "increased" and "decreased" in the manuscript with regard to differences in patterns should instead be "higher" or "lower" proportion). Third, the speech task (autobiographical account) is sufficiently open that patterns of pronoun use may reflect differences in topics or other elements of the narratives rather than eventual cognitive status. It could be argued that these limitations significantly diminish the value of the data.

Nonetheless, I find the value of the paper to be the hypotheses it raises around perspective-taking as indexed by the pronoun use in elaborative and evaluative contexts of the discourse. This seems well beyond the scope of what this study can address but it suggests a cross-linguistic study might be designed to test it out by exploiting the linguistic and referential structures of different languages.

Author Response

Revisions due to comments of Review 2

  • the sample size is very small, and the results of the many comparisons must be viewed with caution.

We agree and revised the ms at various points..

  • the speech samples are taken at only one point in time, prohibiting any speculation of the evolution of pronoun pattern use in cognitive decline

We agree and replaced “decline” by “problems” in the relevant cases.

  • (note that the use of "increased" and "decreased" in the manuscript with regard to differences in patterns should instead be "higher" or "lower" proportion).

We tried to change as suggested. However, in many cases avoiding in/decreased would require complex paraphrasing. We therefore, added footnote 1: For the sake of simplicity, we will speak of "increased" and "decreased" use of pronouns in AD-subjects. It is used in the sense of "more frequently/less frequently than in healthy aging controls". We agree with one reviewer's objection that in our study, as in most other studies on pronoun use, intra-subjective developments have not been investigated.

  • the speech task (autobiographical account) is sufficiently open that patterns of pronoun use may reflect differences in topics or other elements of the narratives rather than eventual cognitive status. It could be argued that these limitations significantly diminish the value of the data.

We added footnote 5: One reviewer expressed concern that patterns of pronoun use may vary due to individual differences in topics or other elements of the interviews and that these limitations significantly diminish the value of the data. We cannot deny individual variation in pronoun use, of course. However, variation in topics is restricted by the fact that all interviews show the same order of questions. With all participants, the main bulk of data stems from reports on their childhood family.
